# Ragulator and GATOR1 complexes promote fission yeast growth by attenuating TOR complex 1 through Rag GTPases

**Kim Hou Chia[1†], Tomoyuki Fukuda[1,2†]\*, Fajar Sofyantoro[1,3], Takato Matsuda[1], Takamitsu Amai[1], Kazuhiro Shiozaki[1,4]\***

[1]Graduate School of Biological Sciences, Nara Institute of Science and Technology, Nara, Japan; [2]Department of Cellular Physiology, Graduate School of Medical and Dental Sciences, Niigata University, Niigata, Japan; [3]Department of Animal Physiology, Faculty of Biology, Universitas Gadjah Mada, Yogyakarta, Indonesia; [4]Department of Microbiology and Molecular Genetics, University of California, Davis, Davis, United States

**Abstract** TOR complex 1 (TORC1) is an evolutionarily conserved protein kinase complex that promotes cellular macromolecular synthesis and suppresses autophagy. Amino-acid-induced activation of mammalian TORC1 is initiated by its recruitment to the RagA/B-RagC/D GTPase heterodimer, which is anchored to lysosomal membranes through the Ragulator complex. We have identified in the model organism *Schizosaccharomyces pombe* a Ragulator-like complex that tethers the Gtr1-Gtr2 Rag heterodimer to the membranes of vacuoles, the lysosome equivalent in yeasts. Unexpectedly, the Ragulator-Rag complex is not required for the vacuolar targeting of TORC1, but the complex plays a crucial role in attenuating TORC1 activity independently of the Tsc1-Tsc2 complex, a known negative regulator of TORC1 signaling. The GATOR1 complex, which functions as Gtr1 GAP, is essential for the TORC1 attenuation by the Ragulator-Rag complex, suggesting that Gtr1$^{GDP}$-Gtr2 on vacuolar membranes moderates TORC1 signaling for optimal cellular response to nutrients.
DOI: https://doi.org/10.7554/eLife.30880.001

**\*For correspondence:**
tfukuda@med.niigata-u.ac.jp (TF);
kaz@bs.naist.jp (KS)

[†]These authors contributed equally to this work

## Introduction

Target of rapamycin (TOR) is serine/threonine-specific protein kinase highly conserved among eukaryotes from yeast to humans. By forming two distinct protein complexes known as TOR complex 1 (TORC1) and TOR complex 2 (TORC2), mammalian TOR (mTOR) and its counterparts in diverse eukaryotes control cell growth, proliferation and metabolisms in response to various intra- and extracellular signals (reviewed in [*Wullschleger et al., 2006*; *Laplante and Sabatini, 2012*]). Mammalian TORC1 (mTORC1) signaling has been attracting much attention, partly because its inhibition by rapamycin is effective in suppressing growth of some human cancers (*Guertin and Sabatini, 2007*; *Shimobayashi and Hall, 2016*). mTORC1 promotes cellular growth through the regulation of both anabolic and catabolic processes; amino acids serve as stimulatory signals that activate mTORC1, which phosphorylates S6K1 and 4E-BP1 to increase protein synthesis (*Hara et al., 1998*). Activated mTORC1 also phosphorylates ULK1, resulting in suppression of autophagy (*Ganley et al., 2009*; *Jung et al., 2009*).

The molecular mechanism that mediates mTORC1 activation in response to amino acid stimuli has recently been a focus of very active research. In the currently prevailing model, a critical step in

amino-acid-induced activation of mTORC1 is its translocation to the surface of lysosomes, where the small GTPase Rheb (*Inoki et al., 2003*; *Tee et al., 2003*) triggers mTORC1 activation. Recruitment of mTORC1 to lysosomes is mediated by the Rag GTPase heterodimer that is anchored to lysosomal membranes through a protein complex called Ragulator (*Kim et al., 2008*; *Sancak et al., 2008*; *Sancak et al., 2010*). The Rag heterodimer consists of RagA or RagB bound to either RagC or RagD, and it has been proposed that guanine nucleotide loading to RagA/B is regulated in response to amino acid signals; the Ragulator complex, which is composed of LAMTOR1/p18, LAMTOR2/p14, LAMTOR3/MP1, LAMTOR4/C7orf59 and LAMTOR5/HBXIP, functions as a guanine nucleotide exchange factor (GEF) for RagA/B (*Bar-Peled et al., 2012*). In the presence of amino acid stimuli, RagA/B are in the GTP-bound form and the Rag heterodimer can interact with the *raptor* subunit of mTORC1 for lysosomal recruitment of mTORC1 (*Sancak et al., 2008*). On the other hand, the GATOR1 complex composed of DEPDC5, Nprl2, and Nprl3 has GTPase-activating protein (GAP) activity toward RagA/B, whose GDP-bound form induces the release of mTORC1 from lysosomes in the absence of amino acids (*Bar-Peled et al., 2013*). Without functional GATOR1, mTORC1 signaling is resistant to amino acid starvation and, interestingly, inactivating mutations to the GATOR1 components have been found in human cancers. GATOR1 forms a larger GATOR complex together with GATOR2, which comprises WDR24, WDR59, MIOS, SEH1L and SEC13, and gene knockdown experiments suggest that the GATOR2 subcomplex negatively regulates the GAP activity of GATOR1 (*Bar-Peled et al., 2013*). It should be noted, however, that further intricacy may remain to be added to the model described above. There is a discrepant report that guanine nucleotide loading to the Rag GTPases is not responsive to amino acids (*Oshiro et al., 2014*). RagA/B-independent activation of mTORC1 has also been observed (*Kim et al., 2014*; *Jewell et al., 2015*).

The mTORC1 regulators enumerated above are conserved also in unicellular eukaryotes. The Gtr1-Gtr2 GTPase heterodimer (*Nakashima et al., 1999*) is the budding yeast counterpart of RagA/B-RagC/D, interacting with the Ego1-Ego2-Ego3 complex that tethers Gtr1-Gtr2 to the membrane of the vacuole, a lysosome-like organelle (*Dubouloz et al., 2005*; *Gao and Kaiser, 2006*; *Powis et al., 2015*; *Kira et al., 2016*). The Ego ternary complex is likely to be the yeast equivalent of mammalian Ragulator, although their constituents share little sequence homology (*Kogan et al., 2010*). The yeast SEACIT and SEACAT complexes apparently correspond to mammalian GATOR1 and GATOR2, respectively (*Neklesa and Davis, 2009*; *Dokudovskaya et al., 2011*; *Wu et al., 2011*; *Panchaud et al., 2013a*; *Panchaud et al., 2013b*; *Kira et al., 2014*). However, these TORC1 regulators in budding yeast might function differently from those in mammals, because the Rheb GTPase, the primary mTORC1 activator, is not part of the TORC1 signaling pathway in budding yeast (*Urano et al., 2000*).

On the other hand, a Rheb ortholog called Rhb1 is an essential activator of TORC1 in the fission yeast *Schizosaccharomyces pombe*, another model eukaryote distantly related to budding yeast (*Mach et al., 2000*; *Urano et al., 2005*; *Uritani et al., 2006*). Like mammalian Rheb, Rhb1 is under the regulation of the Tsc1-Tsc2 complex that functions as GAP for Rhb1 (*Matsumoto et al., 2002*; *van Slegtenhorst et al., 2004*; *Murai et al., 2009*). Thus, *S. pombe* is expected to serve as an excellent experimental system to explore the TORC1 regulatory mechanisms that are conserved also in mammals. Fission yeast TORC1 is composed of the Tor2 kinase associated with the regulatory subunits Mip1 and Wat1, which are orthologous to mammalian *raptor* and mLST8, respectively (*Álvarez and Moreno, 2006*; *Matsuo et al., 2007*; *Hayashi et al., 2007*). The heterodimeric Rag GTPases Gtr1-Gtr2 are also implicated in *S. pombe* TORC1 regulation, although their exact role is ambiguous because of contradictory reports of their mutant phenotypes (*Valbuena et al., 2012*; *Ma et al., 2013*; *Laor et al., 2014*; *Ma et al., 2016*). In this study, we have identified Ragulator- and GATOR1-like complexes in fission yeast, which regulate the cellular localization and nucleotide-binding state of Gtr1-Gtr2, respectively, as has been found with their mammalian counterparts. Unexpectedly, however, these conserved regulatory machineries are all required to attenuate TORC1 activity, and mutants lacking any of them show severe growth defects due to deregulated TORC1 activation. Our data collectively suggest that TORC1 activation in *S. pombe* does not require the Rag-like GTPases and that they rather play an important role in moderating TORC1 activity on vacuolar membranes for optimal cellular response to nutrients.

## Results

### Identification of proteins interacting with Rag GTPases in fission yeast

Aiming to identify proteins that physically interact with the Rag-family GTPases in fission yeast, affinity purification of the Gtr1-Gtr2 heterodimer was performed using strains that express Gtr1 and Gtr2 from their chromosomal loci as fusions with different epitope tags, FLAG and *myc*. Two successive immunoprecipitation procedures using anti-FLAG and anti-*myc* antibodies to collect the Gtr1-Gtr2 heterodimer complex were followed by mass spectrometry, which identified four co-purified proteins encoded by open-reading frames SPBC29A10.17, SPBC1778.05c, SPAC222.19, and SPAC23D3.16 in the *S. pombe* genome database (*Figure 1A and B*). The protein encoded by SPBC1778.05c has been named Lam2 because of its sequence similarity to the mammalian Ragulator subunit LAMTOR2 (*Ma et al., 2016*). On the other hand, the protein products of SPBC29A10.17, SPAC222.19 and SPAC23D3.16 show no apparent sequence homology to any known proteins. However, they form a complex with Lam2, and the predicted secondary structure of the SPAC222.19 protein resembles the αββαββα structure of the roadblock domain (*Koonin and Aravind, 2000*), which has been identified among the human Ragulator components and the Ego proteins in budding yeast (*Bar-Peled et al., 2012*) (*Figure 1—figure supplement 1A*); therefore, we refer to those proteins as Lam1 (SPBC29A10.17), Lam3 (SPAC222.19), and Lam4 (SPAC23D3.16). Their physical interaction with the Gtr1-Gtr2 GTPases was further confirmed by co-immunoprecipitation experiments (*Figure 1C*). In addition, immunoprecipitating any one of the Lam proteins resulted in co-purification of the other three Lam proteins (*Figure 1—figure supplement 1B–E*), confirming complex formation among Lam1, Lam2, Lam3 and Lam4.

Gene disruption analyses found that the null mutation of the *lam1*, *lam2*, *lam3* and *lam4* exhibit very similar growth defects on rich yeast extract (YES) medium (*Figure 1D*). We also noticed that their growth phenotypes were comparable to those of the cells lacking Gtr1 or Gtr2. Moreover, the defective phenotypes of the *lamΔ gtr1Δ* and *lamΔ gtr2Δ* double mutants are indistinguishable from those of the respective single mutants, indicating that the *lamΔ* and *gtrΔ* phenotypes are not additive (*Figure 1D and E*). Together, these biochemical and genetic data suggest that the Gtr1-Gtr2 GTPases and the Lam1~4 proteins form a complex and function together to promote cell growth.

### The Lam protein complex tethers Gtr1-Gtr2 GTPases to vacuolar membranes

Similar to the lysosomal localization of the mammalian Rag GTPases, the Gtr1-Gtr2 heterodimer localizes to membranes of the vacuole, the lysosome equivalent in yeasts (*Valbuena et al., 2012*). To examine whether the Lam proteins are also localized to vacuolar membranes, we constructed strains in which the GFP-encoding sequence was inserted at the 3′ end of the chromosomal *lam1*[+], *lam2*[+], *lam3*[+] and *lam4*[+] open-reading frames. By fluorescence microscopy, all four Lam-GFP fusion proteins were detected on vacuolar membranes labeled by the FM4-64 dye (*Figure 2A*). The Lam proteins were also detectable on vacuoles even in strains lacking Gtr1 and Gtr2 (*Figure 2—figure supplement 1*), indicating that the vacuolar targeting of the Lam proteins does not require the Gtr1-Gtr2 GTPases. We next assessed the interdependency among the Lam proteins for their vacuolar localization. In the *lam1Δ* mutant, Lam2, Lam3 and Lam4 diffused throughout the cytoplasm, while Lam1 was observed on vacuoles even in *lam2Δ*, *lam3Δ* and *lam4Δ* cells (*Figure 2B*, *Figure 2—figure supplement 2*); thus, Lam1 appears to be indispensable for anchoring the Lam complex to vacuoles. On the other hand, the vacuolar localization of Lam2 and Lam3 is also dependent on Lam4 but not vice versa (*Figure 2B*, *Figure 2—figure supplement 2*), implying that Lam4 mediates association of Lam2 and Lam3 with Lam1 anchored to vacuoles. We also observed that Lam2 and Lam3 are mutually dependent for their vacuolar localization.

Consistent with the inferred role of Lam1 as a vacuolar membrane anchor of the Lam complex, its N-terminal region has potential myristoylation and palmitoylation sites (*Figure 2C*) similar to those in the Ragulator component LAMTOR1 and the budding yeast Ego1 protein (*Kogan et al., 2010*). Moreover, the vacuole/lysosome localization signal sequence composed of acidic residues followed by di-leucine (*Darsow et al., 1998*) is also conserved within the N-terminal regions of Lam1, LAMTOR1, and Ego1 (*Figure 2C*, underlines). We found that simultaneous alanine substitutions of both myristorylation and palmitoylation sites in Lam1 resulted in its dispersal throughout the cytoplasm

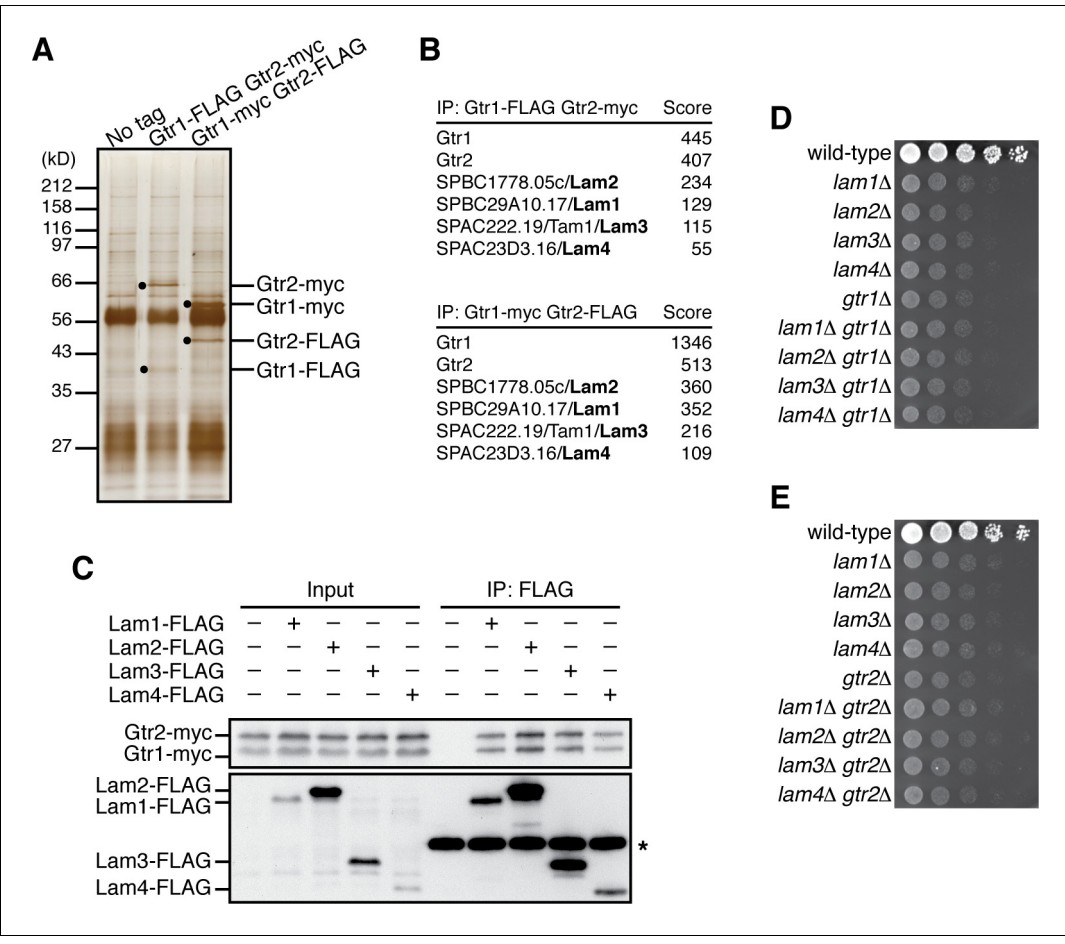

**Figure 1.** Identification of the Lam proteins that interact with the Gtr1-Gtr2 Rag GTPase heterodimer. (**A**) Affinity-purification of the Gtr1-Gtr2 heterodimer from *S. pombe*. The heterodimer was purified from the cell lysate of *gtr1:FLAG gtr2:myc* and *gtr1:myc gtr2:FLAG* strains by two successive immunoprecipitation procedures using anti-FLAG and anti-*myc* antibodies, and resolved on SDS-PAGE followed by silver staining. A wild-type strain that expresses the untagged Gtr proteins ('No tag') was used as a negative control. The protein bands corresponding to the tagged Gtr1 and Gtr2 are indicated by black dots. (**B**) Mass spectrometric analyses of proteins co-purified with the Gtr1-Gtr2 heterodimer in the experiments shown in (**A**). Two or more peptides were identified for each protein listed in the tables. The sum of peptide scores that exceed the 95% confidence level (p<0.05) is also shown for each of the identified proteins. (**C**) Confirmed physical interactions between the Lam proteins and Gtr1-Gtr2. Crude lysate ('Input') was prepared from *gtr1:myc gtr2:myc* strains expressing one of the Lam proteins with the FLAG tag, and anti-FLAG immunoprecipitates ('IP:FLAG') were analyzed by immunoblotting. Immunoglobulin in the immunoprecipitates is asterisked. (**D–E**) The *lam* and *gtr* genes function in the same pathway. The indicated single and double mutants as well as a wild-type strain were grown in EMM liquid medium and their serial dilutions were spotted onto YES agar medium for a growth assay at 30˚C.

DOI: https://doi.org/10.7554/eLife.30880.002

The following figure supplement is available for figure 1:

**Figure supplement 1.** The *S. pombe* Lam proteins form a complex equivalent of mammalian Ragulator and the Ego ternary complex in budding yeast.

DOI: https://doi.org/10.7554/eLife.30880.003

(*Figure 2D* 'lam1GCC-AAA'). On the other hand, Lam1 was distributed to both vacuolar and plasma membranes when the conserved di-leucine motif of Leu-21 and Leu-22 was replaced by two alanine residues (*lam1LL-AA*). These observations suggest that specific targeting of Lam1 to vacuolar membranes requires its N-terminal lipid modifications as well as the di-leucine motif.

In addition to the detectable sequence similarity of Lam2 to mammalian LAMTOR2, the aforementioned characteristics conserved among Lam1, LAMTOR1 and Ego1 suggest that the Lam

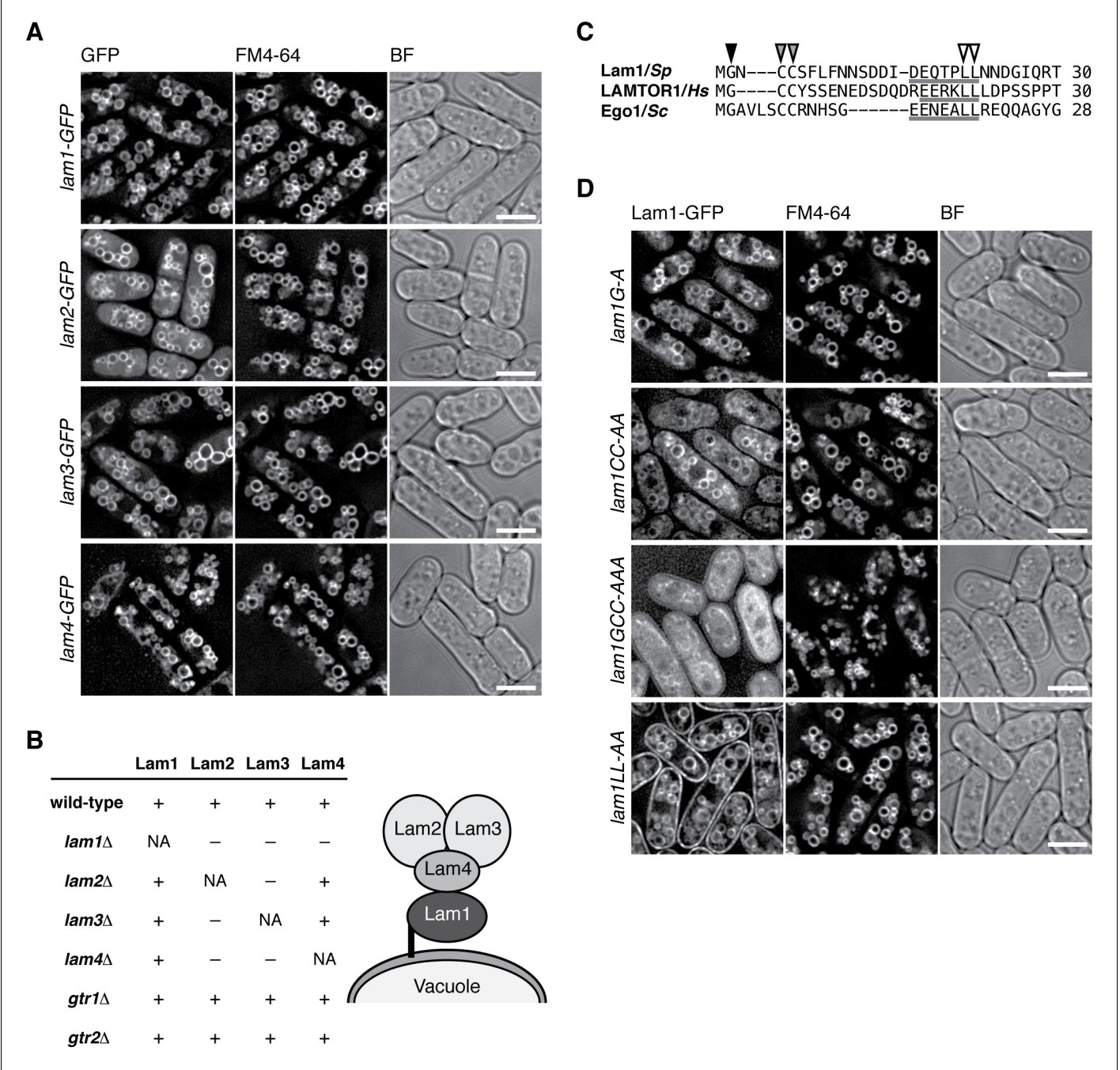

**Figure 2.** The Lam protein complex is localized to vacuolar membranes. (**A**) The chromosomal *lam* genes were tagged with the GFP sequence and the strains were grown in EMM at 30°C for microscopy, with vacuolar membranes visualized by the fluorescent dye FM4-64. Z-axial images were collected and mid-section images after deconvolution are shown. BF, bright-field image. Bars, 5 µm. (**B**) Interdependent vacuolar localization of the Lam proteins. See *Figures 2A* and *Figure 2—figure supplements 1 and 2* for representative microscopy images. '+', localization to vacuolar membranes; '–', defused in the cytosol. Predicted architecture of the Lam complex on the vacuolar surface is also shown as a schematic diagram. (**C**) Sequence alignment of the N-terminal regions of *S. pombe* Lam1 (UniProtKB ID: C6Y4C6), human LAMTOR1 (Q6IAA8) and Ego1 in *Saccharomyces cerevisiae* (Q02205). The conserved Gly (black arrowhead) and Cys (gray arrowheads) residues are potential sites for myristoylation and palmitoylation, respectively. Putative vacuole/lysosome localization signal sequences composed of acidic residues followed by a di-leucine motif (open arrowheads) are underlined. (**D**) The sequence motifs shown in (**C**) are important for the vacuolar localization of Lam1. The *lam1:GFP* strains with alanine-substitutions at the potential myristoylation site (G–A), palmitoylation sites (CC-AA), both of them (GCC-AAA) or the di-leucine (LL-AA) motif were analyzed as in (**A**). Bars, 5 µm.

DOI: https://doi.org/10.7554/eLife.30880.004

*Figure 2 continued on next page*

*Figure 2 continued*

The following figure supplements are available for figure 2:

**Figure supplement 1.** Vacuolar localization of the Lam proteins does not require Gtr1-Gtr2.
DOI: https://doi.org/10.7554/eLife.30880.005
**Figure supplement 2.** Interdependent vacuolar localization of the Lam proteins.
DOI: https://doi.org/10.7554/eLife.30880.006

protein complex is a Ragulator equivalent in fission yeast. The vacuolar localization of the Lam proteins (*Figure 2A*) and their physical association with the Gtr1-Gtr2 GTPases (*Figure 1*) are also consistent with such a possibility. Therefore, we investigated whether the Lam protein complex is responsible for the localization of the Gtr1-Gtr2 GTPases to vacuolar membranes, as is the case with mammalian Ragulator for the lysosomal localization of the Rag GTPases (*Sancak et al., 2010*). Indeed, the vacuolar localization of Gtr1-Gtr2 is impaired in the absence of any one of the Lam proteins (*Figure 3A*, *Figure 3—figure supplement 1A*), indicating that the intact Lam complex is required for the recruitment of Gtr1-Gtr2 to vacuoles. Heterodimer formation between Gtr1 and Gtr2 is also essential for their localization to vacuoles, because these proteins diffused throughout the cytoplasm in the absence of the other (*Figure 3A* '*gtr2Δ*' and *Figure 3—figure supplement 1A* '*gtr1Δ*'). Intriguingly, artificial targeting of the Lam1 protein to the nucleus by fusing the nuclear localization signal sequence of the SV40 large T-antigen resulted in the translocation of Gtr1-Gtr2 to the nucleus (*Figure 3B*). Thus, the Lam protein complex appears to be a key determinant of the cellular localization of Gtr1-Gtr2, serving as a Ragulator-like complex that recruits the fission yeast Rag GTPases to vacuolar membranes.

In a currently prevailing model, recruitment of mTORC1 to lysosomal membranes by the Ragulator-Rag complex is a key step in mTORC1 activation by the Rheb GTPase (*Sancak et al., 2010*). Vacuolar localization of fission yeast TORC1 was previously reported (*Valbuena et al., 2012*), which we confirmed by constructing a strain that expresses GFP-tagged Mip1, a *raptor* ortholog, from the chromosomal *mip1*[+] locus (*Figure 3C* 'wild-type'). To our surprise, Mip1-GFP was observed mostly on vacuolar membranes even in *lam1Δ*, *gtr1Δ* and *gtr2Δ* mutant cells. In addition, Mip1 persisted on vacuoles even when the Gtr1-Gtr2 GTPases were translocated to the nucleus by NLS-fused Lam1 (*Figure 3B*, bottom panels). These observations indicate that fission yeast TORC1 can localize to vacuolar membranes in a manner independent of the Ragulator-Rag complex.

## Identification of a GATOR1-like protein complex in fission yeast

The GATOR1 complex, which consists of DEPDC5, Nprl2 and Nprl3, has GAP activity for RagA/B, and its inactivating mutations in cancers result in mTORC1 hyperactivation (*Bar-Peled et al., 2013*). Nprl2 has a fission yeast ortholog, Npr2, and genetic analysis suggested that Npr2 negatively regulates TORC1 also in fission yeast (*Ma et al., 2013*). To determine whether Npr2 is part of a GATOR1-like complex, Npr2 was affinity-purified from the lysate of a strain expressing FLAG-tagged Npr2 (*Figure 4A*), and the co-purified proteins were analyzed by mass spectrometry (*Figure 4B*). The identified Npr2-interacting proteins include Iml1 (SPBC26H8.04c) and Npr3 (SPBC543.04), which show detectable sequence similarity to the mammalian GATOR1 components DEPDC5 and Nprl3, respectively (*Figure 4—figure supplement 1*). Physical association of Npr2, Iml1 and Npr3 was also confirmed by pair-wise immunoprecipitation experiments (*Figure 4—figure supplement 2A–C*), suggesting that the three proteins form a protein complex equivalent to mammalian GATOR1. Strains lacking the Iml1, Npr2 or Npr3-coding genes exhibited similar growth defects on YES agar medium (*Figure 4C*). In addition, the double- and triple-deletion mutants of these genes showed defective phenotypes comparable to the respective single mutants, consistent with the notion that Iml1, Npr2 and Npr3 function as a complex. We also noticed that the amino acid sequences of Sea4, Sec13, Sea3, and Seh1 detected in the FLAG-Npr2 immunoprecipitates (*Figure 4B*) show similarity to those of the mammalian GATOR2 components MIOS, Sec13, WDR59, and SEH1, respectively (http://www.pombase.org; our unpublished results), implying conservation of the entire GATOR complex in *S. pombe*.

Fluorescence microscopy of fission yeast strains whose chromosomal *iml1*[+] gene was tagged with the GFP sequence revealed Iml1 localization to vacuolar membranes in wild-type cells (*Figure 4D*) as

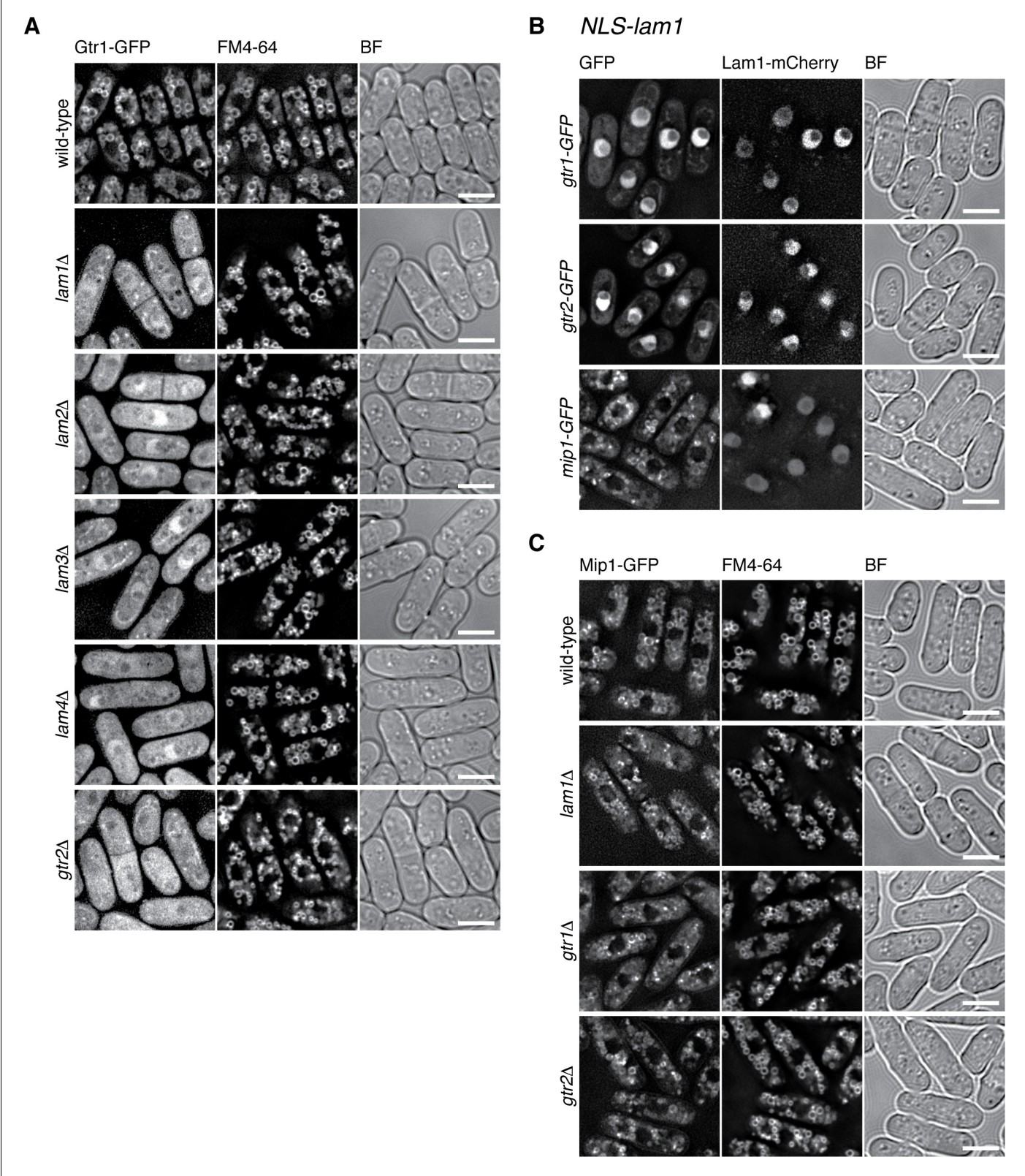

**Figure 3.** The Lam complex is the Ragulator equivalent that tethers Gtr1-Gtr2 to vacuolar membranes. (**A**) The Lam proteins and Gtr2 are essential for the vacuolar localization of Gtr1. Wild-type and the indicated mutant strains expressing GFP-tagged Gtr1 from its chromosomal locus were analyzed by fluorescence microscopy as in *Figure 2A*. Bars, 5 μm. (**B**) Artificial nuclear targeting of Lam1 translocates Gtr1-Gtr2, but not TORC1, to the nucleus.
*Figure 3 continued on next page*

*Figure 3 continued*

Lam1GCC-AAA that cannot localize to vacuoles (*Figure 2D*) was fused to the SV40 nuclear localization signal (NLS) and mCherry sequences, and expressed in the *gtr1:GFP*, *gtr2:GFP* and *mip1:GFP* strains for fluorescence microscopy. Bars, 5 μm. (**C**) The Ragulator-Rag complex is not required for vacuolar localization of TORC1. Wild-type and the indicated mutant strains that express the Mip1 subunit of TORC1 with the GFP tag were analyzed by fluorescence microscopy. Bars, 5 μm.

DOI: https://doi.org/10.7554/eLife.30880.007

The following figure supplement is available for figure 3:

**Figure supplement 1.** The Lam complex is important for the vacuolar localization, but not the stability, of the Gtr1-Gtr2 proteins.

DOI: https://doi.org/10.7554/eLife.30880.008

well as in strains lacking Npr2 or Npr3 (*Figure 4—figure supplement 2D*). In addition, the vacuolar localization of Iml1 did not appear to require the Gtr1-Gtr2 GTPases and the Ragulator-like complex on vacuolar membranes (*Figure 4D*, *Figure 4—figure supplement 2E*). Furthermore, the Iml1-Npr2-Npr3 complex was found to be dispensable for targeting Gtr1-Gtr2 to vacuoles (*Figure 4E*, *Figure 4—figure supplement 2F–H*). Thus, the Ragulator-Rag complex and the GATOR1 complex in fission yeast are both localized to vacuolar membranes, but in a manner independent of each other.

## GATOR1-like complex and the Gtr1-Gtr2 GTPases promote cell growth by attenuating TORC1 signaling

Loss of functional GATOR1 has been identified in cancer cell lines, implying that hyperactive mTORC1 in the absence of GATOR1 contributes to cell proliferation (*Bar-Peled et al., 2013*). Therefore, the growth defect observed in the absence of the GATOR1-like complex in fission yeast (*Figure 4C*) was an unexpected result, and further genetic analysis was conducted to determine the relationship between the Iml1-Npr2-Npr3 complex and the Gtr1-Gtr2 Rag GTPases. The compromised growth phenotypes of the *iml1Δ*, *npr2Δ* and *npr3Δ* mutants on YES medium were found to be comparable to those of the *gtr1Δ* and *gtr2Δ* strains, and the double mutant analyses indicated that the GATOR1 and Rag GTPase mutant phenotypes are not additive (*Figure 5A and B*). Thus, *S. pombe* GATOR1 appears to contribute to cell growth in the same pathway as the Gtr1-Gtr2 GTPases. To further examine the possibility that the GATOR1 complex regulates the nucleotide-binding state of Gtr1, we constructed strains whose *gtr1+* locus was replaced with *gtr1Q61L* or *gtr1S16N* that are expected to express constitutively GTP- and GDP-bound Gtr1, respectively (*Nakashima et al., 1999*; *Valbuena et al., 2012*). Cells expressing GTP-locked Gtr1Q61L showed a severe growth defect similar to the *gtr1Δ* mutant, while the growth of the *gtr1S16N* mutant was comparable to that of wild-type cells, implying an important function of the GDP-bound form of Gtr1 (*Figure 5C*). On the other hand, the equivalent mutations to Gtr2 did not significantly affect the phenotype. The *gtr2Δ* mutation, however, caused a growth defect indistinguishable from that of the *gtr1Δ* mutant and was found to be epistatic to any of the *gtr1* mutations including *gtr1S16N* (*Figure 5D*), indicating the essential role of Gtr2 for the Gtr1 function.

If, like mammalian GATOR1 (*Bar-Peled et al., 2013*), the Iml1-Npr2-Npr3 complex functions as GAP for Gtr1, GTP-bound Gtr1 should accumulate in the absence of the Iml1-Npr2-Npr3 complex. Indeed, the GTP-locked *gtr1Q61L* mutation and the GATOR1 mutations caused very similar growth phenotype, and their defects were not additive in the double mutants (*Figure 5E*). On the contrary, the GDP-locked *gtr1S16N* mutation suppressed the phenotype of the GATOR1-defective mutants (*Figure 5F*). These results support the notion that GATOR1 functions as GAP for Gtr1 and that loss of functional GATOR1 is complemented by expression of the GDP-bound form of Gtr1. Consistently, *iml1R854A*, a mutation to the 'arginine finger' conserved within the GAP domain of Iml1 orthologs (*Figure 4—figure supplement 1A*) (*Panchaud et al., 2013a*), caused a growth defect similar to those of the *lam1Δ* and *gtr1Q61L* strains (*Figure 4—figure supplement 2I*).

The observations above strongly suggested an important function of GDP-bound Gtr1 (Gtr1$^{GDP}$) in growth of *S. pombe*, and we hypothesized that optimal growth of fission yeast under nutrient-rich conditions, such as yeast extract medium, might require attenuation of TORC1 by the heterodimer of Gtr1$^{GDP}$ and Gtr2. As expected, the TORC1 inhibitors rapamycin and caffeine (*Takahara and Maeda, 2012*; *Rallis et al., 2013*) suppressed the defective growth phenotypes of the strains lacking Gtr1-Gtr2 or the GATOR1 components as well as that of the GTP-locked *gtr1Q61L* mutant (*Figure 5G*, *Figure 5—figure supplement 1A*). Similarly, temperature-sensitive hypomorphic *tor2*

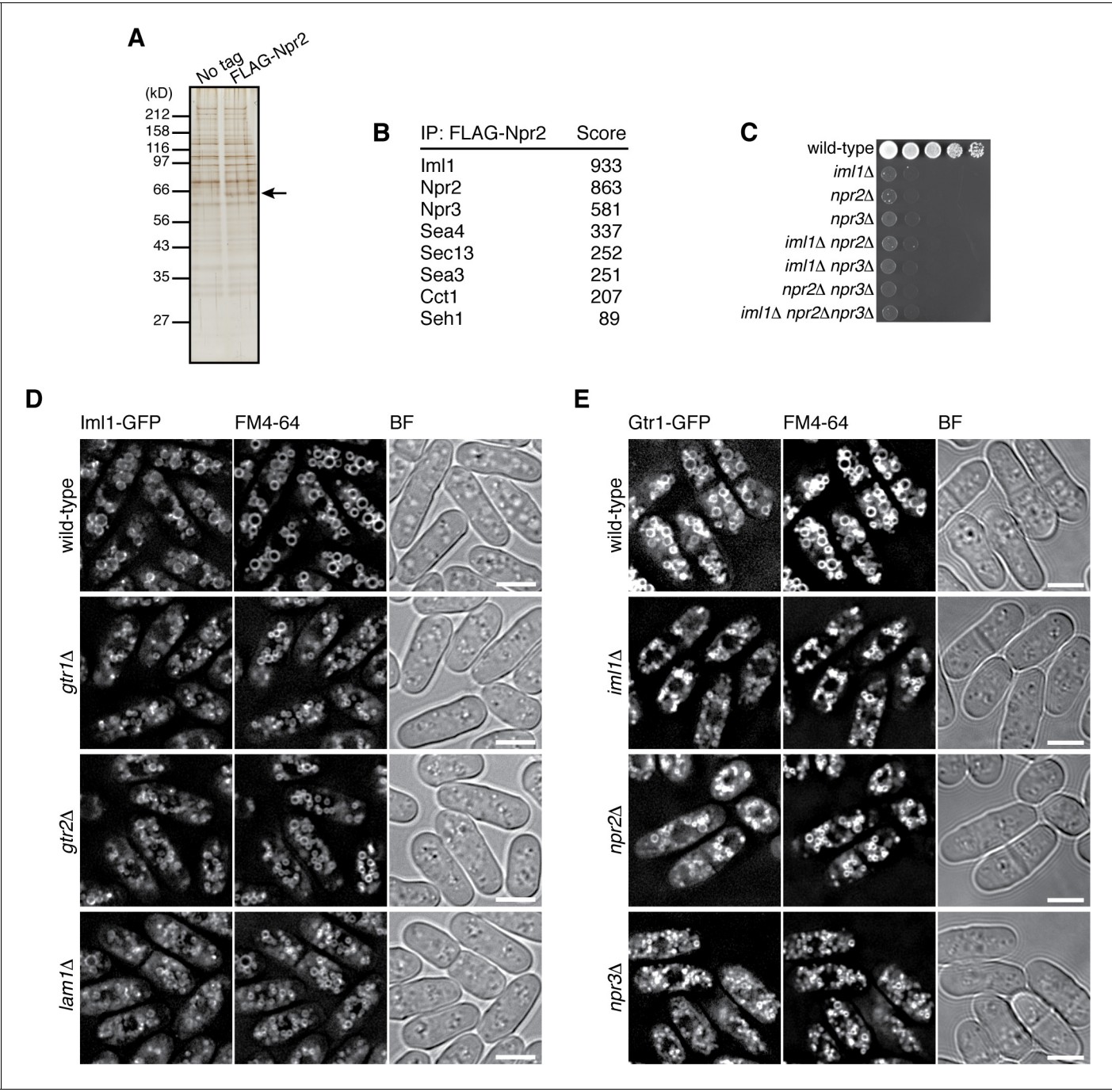

**Figure 4.** Identification of the fission yeast GATOR1 complex. (**A**) Affinity purification of FLAG-tagged Npr2 and its associating proteins. Cells expressing FLAG-Npr2 were grown in YES medium and their lysate was subjected to anti-FLAG immunoprecipitation. The eluates were resolved on SDS-PAGE followed by silver staining. A wild-type strain that expresses untagged Npr2 ('No tag') was used as a negative control. The protein band corresponding to FLAG-Npr2 is indicated by an arrow. (**B**) Mass spectrometric analyses of proteins co-purified with FLAG-Npr2 in the experiment shown in (**A**) identified *S. pombe* orthologs of the GATOR1 (Iml1 and Npr3) and GATOR2 (Seh1, Sea3, Sea4 and Sec13) subunits. Two or more peptides were identified for each protein listed in the table. The sum of peptide scores that exceed the 95% confidence level (p<0.05) is also shown for each of the identified proteins. (**C**) Growth defects of the mutants lacking the GATOR1 subunits. The indicated single, double and triple mutants as well as a wild-type strain were grown in EMM liquid medium and their serial dilutions were spotted onto YES agar medium at 30°C. (**D**) Vacuolar localization of GATOR1 is independent of the Ragulator-Rag complex. The *iml1:GFP* locus was introduced to the wild type and the mutants lacking Gtr1-Gtr2 or Lam1 and the localization of Iml1-GFP was analyzed by fluorescence microscopy. Bars, 5 μm. (**E**) Vacuolar localization of the Gtr1 GTPase is independent of GATOR1. Wild-type and GATOR1 mutant strains expressing GFP-tagged Gtr1 from its chromosomal locus were analyzed by fluorescence microscopy. Bars, 5 μm.

*Figure 4 continued on next page*

*Figure 4 continued*

DOI: https://doi.org/10.7554/eLife.30880.009

The following figure supplements are available for figure 4:

**Figure supplement 1.** Fission yeast Iml1 and Npr3 are orthologous to the mammalian GATOR1 components, DEPDC5 and Nprl3, respectively.

DOI: https://doi.org/10.7554/eLife.30880.010

**Figure supplement 2.** Expression and vacuolar localization of Gtr1-Gtr2 are not regulated by GATOR1.

DOI: https://doi.org/10.7554/eLife.30880.011

alleles, *tor2-13* and *tor2-287* (*Uritani et al., 2006*; *Hayashi et al., 2007*; *Ikai et al., 2011*), also complemented the loss of Gtr1-Gtr2, GATOR1 or Ragulator (*Figure 5H–K*, *Figure 5—figure supplement 1B and C*), corroborating that cells with reduced TORC1 activity can grow in the absence of Gtr1-Gtr2 and GATOR1. Indeed, absence of Gtr1-Gtr2, GATOR1 or Ragulator allowed *tor2-287* mutant cells to grow even at the restrictive temperature (*Figure 5I and K* and *Figure 5—figure supplement 1C*), suggesting that loss of Gtr1$^{GDP}$-Gtr2 leads to augmented TORC1 activity in this *tor2* mutant.

Collectively, the above data are consistent with a model where the GATOR1 GAP gives rise to Gtr1$^{GDP}$, which forms heterodimer with Gtr2 to moderate TORC1 activity to a level optimal for cell growth. Because the Tsc1-Tsc2 complex also negatively regulates TORC1 signaling as GAP for the Rhb1 GTPase (*Urano et al., 2005*; *Uritani et al., 2006*; *van Slegtenhorst et al., 2004*; *Nakase et al., 2006*), we examined the relationship of Tsc1-Tsc2 with GATOR1-Gtr1. Without the functional Tsc complex in the *tsc1Δ* or *tsc2Δ* background, both *gtr1Δ* and *iml1Δ* mutations brought about growth defects much severer than those of the single mutants (*Figure 5L and M*), and these synthetic phenotypes were suppressed by rapamycin (*Figure 5—figure supplement 1D*). Thus, it appears that the GATOR1-Gtr signaling axis and the Tsc-Rhb1 pathway independently impinge on TORC1 (*Figure 5N*) and that loss of both negative regulatory mechanisms results in a severe growth defect.

## TORC1 activity is elevated in the absence of GDP-bound Gtr1

The genetic analyses presented above led to a working model that GATOR1 and Gtr1$^{GDP}$-Gtr2 temper TORC1 activation induced by the Rhb1 GTPase (*Figure 5N*); loss of Gtr1$^{GDP}$-Gtr2 from vacuolar membranes results in deregulated TORC1 signaling and growth defects that can be suppressed by rapamycin or Tor2 hypomorphs. In order to further substantiate this model, we next examined how GATOR1 and Gtr1-Gtr2 affect TORC1 activity by monitoring phosphorylation of Psk1, a *S. pombe* S6K1 ortholog. The hydrophobic motif phosphorylation of Psk1 by TORC1 results in reduced electrophoretic mobility of Psk1 and is also detectable using antibodies against phosphorylated S6K1 (*Nakashima et al., 2012*). As previously reported (*Hatano et al., 2015*), phosphorylated Psk1 became undetectable in wild-type cells within 30 min of nitrogen starvation that inactivates TORC1 (*Figure 6A*). In contrast, Psk1 remained phosphorylated until later time points in strains lacking the Tsc1-Tsc2 complex, a known negative regulator of TORC1 signaling, as well as in those carrying the activating mutations of Tor2 and Rhb1, *tor2E2221K* (*Urano et al., 2007*) and *rhb1-DA4* (*Murai et al., 2009*), respectively (*Figure 6B*). Similar delays in TORC1 inactivation were observed in the *gtr1Δ* and *gtr2Δ* mutants (*Figure 6C*), indicating that absence of Gtr1-Gtr2 results in deregulated TORC1 signaling that cannot promptly respond to starvation. The GTP-locked *gtr1Q61L* mutant and *iml1Δ* mutant also exhibited a delay in the nitrogen-starvation response, while no apparent difference was observed between the GDP-locked *gtr1S16N* mutant and the wild type (*Figure 6D and E*). These data are consistent with our model that GATOR1 functions as GAP for Gtr1 and that the Gtr1$^{GDP}$-Gtr2 heterodimer plays a role in restraining TORC1 activation. As expected, the GDP-locked *gtr1S16N* mutation suppressed the delayed starvation response of the *iml1Δ* mutant that lacks the functional GATOR1 complex (*Figure 6F*), consistent with the observed phenotypic complementation (*Figure 5F*). In addition, the parallel TORC1 regulation by the Tsc1-Tsc2 and GATOR1-Gtr pathways (*Figure 5N*) was also observed by monitoring the TORC1 activity; the TORC1 response to nitrogen starvation was further delayed in the *gtr1Δ tsc2Δ* (*Figure 6G*) and *iml1Δ tsc2Δ* double mutants (*Figure 6H*) in comparison to the respective single mutants. We thus examined further the possibility that the GATOR1-Gtr pathway can regulate TORC1 signaling independently of the Rhb1

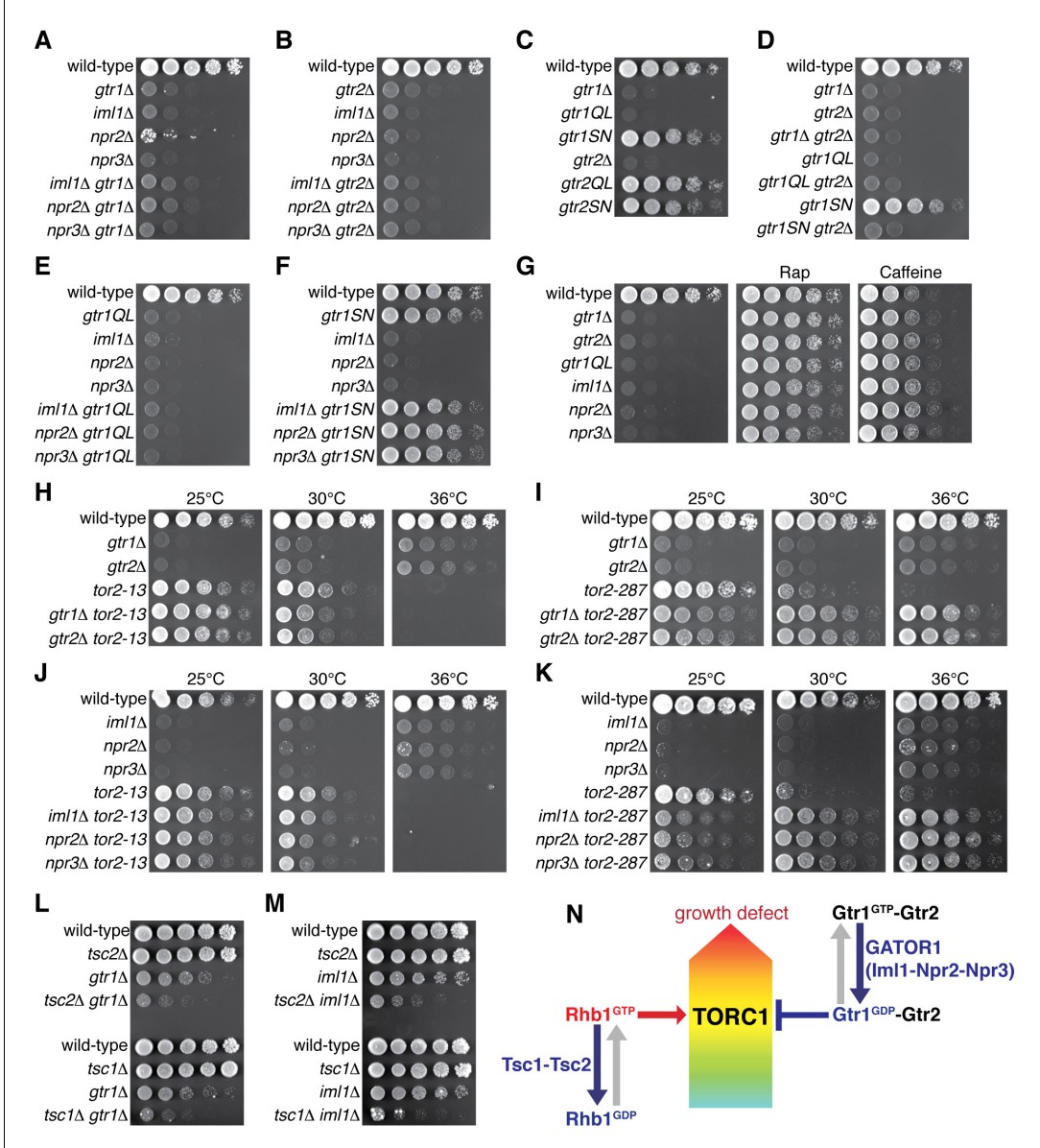

**Figure 5.** Gtr1$^{GDP}$-Gtr2 attenuates TORC1 signaling for normal cell growth. Wild-type and the indicated mutant strains were grown in EMM liquid medium and their serial dilutions were spotted onto YES agar media at 30°C unless indicated otherwise. (**A, B**) GATOR1 and Gtr1-Gtr2 function in the same pathway. Growth of the indicated single and double mutants were compared. (**C, D**) Growth phenotypes of the strains expressing the GTP-locked ('*gtr1QL*' and '*gtr2QL*') and GDP-locked ('*gtr1SN*' and '*gtr2SN*') mutants of the Gtr GTPases. Wild-type and GDP-locked Gtr1 allow normal cell growth only in the presence of Gtr2. (**E, F**) Genetic interactions between GATOR1 and Gtr1-Gtr2. The growth defects caused by the loss of GATOR1 are complemented by expressing the GDP-bound Gtr1S16N. (**G**) TORC1 inhibitors suppress the growth defects of the mutants lacking Gtr1$^{GDP}$-Gtr2. Growth of the indicated strains was tested on YES agar with 100 ng/ml rapamycin (Rap) or 10 mM caffeine. (**H–K**) Temperature-sensitive, hypomorphic mutations of the TOR kinase, *tor2-13* and *tor2-287*, suppress the growth defects of the cells lacking Gtr1-Gtr2 or GATOR1. Growth of the indicated single and double mutants was tested at the permissive (25°C), semi-permissive (30°C) and restrictive (36°C) temperatures of the *tor2* mutants. (**L–M**) The GATOR1-Gtr pathway negatively regulates TORC1 independently of the Tsc complex. The growth defects of the *gtr1Δ* and *iml1Δ* mutants were accentuated in the absence of Tsc1 or Tsc2. (**N**) A working model for the TORC1 regulation by GATOR1 and Gtr1-Gtr2 in fission yeast. GATOR1 composed of Iml1, Npr2 and Npr3 functions as GAP for Gtr1, which forms heterodimer with Gtr2 to moderate TORC1 activation induced by the GTP-bound Rhb1 GTPase. Hyperactivation of TORC1 in the absence of Gtr1$^{GDP}$-Gtr2 results in a growth defect as found in (**A–K**).

DOI: https://doi.org/10.7554/eLife.30880.012

The following source data and figure supplement are available for figure 5:

**Source data 1.** Mutation sites used to generate the GDP-locked mutant forms of RagA/B GTPases.

DOI: https://doi.org/10.7554/eLife.30880.014

*Figure 5 continued on next page*

*Figure 5 continued*

**Figure supplement 1.** Ragulator is important for attenuation of TORC1 signaling in *S. pombe*.
DOI: https://doi.org/10.7554/eLife.30880.013

GTPase. The Rhb1 is an essential activator of TORC1 and the *rhb1Δ* mutant is viable only in the presence of an activated allele of Tor2, such as *tor2E2221K* (*Mach et al., 2000*; *Urano et al., 2007*). Also in the *tor2E2221K* background, the *gtr1Δ* and *iml1Δ* mutations caused significant delays in the TORC1 response to starvation, indicating that TORC1 carrying the *tor2E2221K* mutation is still subject to the regulation by the GATOR1-Gtr pathway (*Figure 6I*). Interestingly, even in the absence of Rhb1, the *gtr1Δ* and *iml1Δ* mutations caused apparent delays in the TORC1 inactivation upon starvation (*Figure 6J*), suggesting that Gtr1$^{GDP}$-Gtr2 can control TORC1 activity in a Rhb1-independent manner, at least in the *tor2E2221K* background.

In the presence of plentiful nutrients, Psk1 appeared to be mostly in the phosphorylated, slow-migrating form and very little unphosphorylated Psk1 was detectable (e.g. *Figure 6A*, middle panel at time 0). On the other hand, the slow-migrating, phosphorylated form of Sck1, which is another known substrate of TORC1 (*Nakashima et al., 2012*), significantly increased in growing *gtr1Δ*, *gtr2Δ* and *iml1Δ* mutant cells, comparing to that in wild-type cells (*Figure 6K*). This observation suggests that absence of GATOR1 or Gtr1-Gtr2 results in elevated TORC1 activity also under nitrogen-replete conditions. In contrast, loss of the Tsc1-Tsc2 complex seems to have little impact on TORC1 activity under the same conditions.

Together, the above results demonstrate hyperactive TORC1 signaling in the absence of GATOR1 or Gtr1$^{GDP}$-Gtr2, consistent with the phenotypic analyses shown in *Figure 5*.

## The growth defect in the absence of Gtr1$^{GDP}$-Gtr2 is partly attributed to impaired amino-acid uptake

Our studies described above showed that loss of Gtr1$^{GDP}$-Gtr2 results in deregulated TORC1 activation that brings about a growth defect in the yeast extract medium. We noticed that the defective phenotype was alleviated by supplementing the medium with ammonium (*Figure 7A*, *Figure 7—figure supplement 1A*), implying that loss of the functional GATOR1-Gtr pathway starves *S. pombe* cells of nitrogen in the standard yeast extract medium. Because TORC1 is known to control cellular amino-acid uptake (*Matsumoto et al., 2002*; *van Slegtenhorst et al., 2004*; *Aspuria and Tamanoi, 2008*; *Liu et al., 2015*), we examined the possibility that the observed growth defect is due to compromised amino-acid uptake. The *gtr*, GATOR1 and Ragulator mutants could grow on EMM minimal medium that contains ammonium as nitrogen source, but little growth of the mutant cells was observed when ammonium was replaced with amino acid (*Figure 7B*, *Figure 7—figure supplement 1B*). Addition of ammonium chloride significantly mitigated their growth defect on the amino-acid media, implying that mutant cells lacking Gtr1$^{GDP}$ or Ragulator can utilize ammonium but not amino acids as nitrogen source. The defective amino-acid uptake of the *gtr*, GATOR1 and Ragulator mutants was also evidenced by their resistance to canavanine and ethionine, toxic analogs of arginine and methionine, respectively (*Figure 7C*, *Figure 7—figure supplement 1C*). As expected, those mutants with amino-acid auxotrophy failed to grow by utilizing exogenous amino acids in the growth medium (*Figure 7—figure supplement 1D and E*).

It has been reported that the Pub1 ubiquitin ligase and its arrestin-related adaptor, Any1, negatively regulate the plasma-membrane localization of amino acid transporters through endocytosis and their retention to Golgi/endosomes (*Nakase et al., 2013*; *Nakashima et al., 2014*). The *pub1Δ* and *any1Δ* mutations partially suppressed the growth defect of the *gtr1Δ* and *iml1Δ* mutants on YES medium and abrogated their resistance to the toxic amino acid analogs, implying ameliorated amino-acid uptake in the double mutants (*Figure 7D*). Indeed, the *pub1Δ* and *any1Δ* mutations failed to improve the growth of *gtr1Δ* and *iml1Δ* cells on EMM, whose nitrogen source is ammonium rather than amino acid. We therefore examined whether the *gtr1Δ* and *iml1Δ* mutations affect the plasma-membrane localization of Cat1, an amino acid transporter known to be under the regulation of Pub1 and Any1 (*Nakashima et al., 2014*). The majority of Cat1 with a fluorescent tag was detected at the plasma membrane in wild-type cells, but the protein was mostly sequestered in the cytoplasm of the *gtr1Δ* and *iml1Δ* mutants (*Figure 7E*). Furthermore, the *pub1Δ* (*Figure 7F*) and

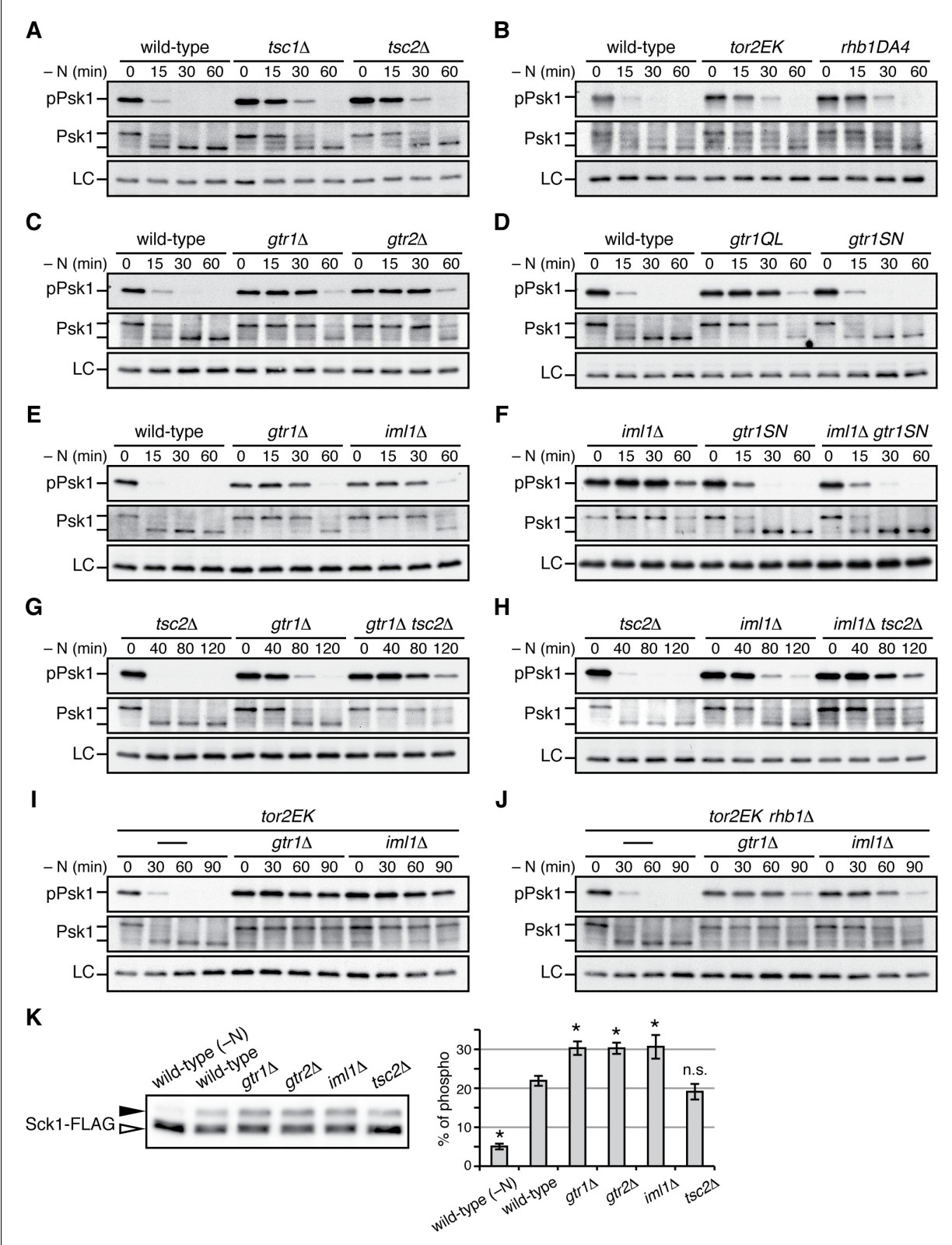

**Figure 6.** Gtr1$^{GDP}$-Gtr2 negatively regulates TORC1 signaling. (**A–J**) TORC1 activity was monitored by immunoblotting to detect the TORC1-dependent phosphorylation of Psk1 ('pPsk1'), along the time course after the indicated strains exponentially growing in EMM at 30°C were shifted to the same medium without nitrogen source. The samples were also probed with anti-Psk1 antibodies ('Psk1') as well as anti-Spc1 MAPK antibodies to control loading ('LC'). Inactivation of TORC1 after nitrogen starvation was delayed in the strains lacking the Tsc complex (**A**) and those carrying the activating

*Figure 6 continued on next page*

*Figure 6 continued*

mutations *tor2EK* or *rhb1DA4* (B). Similarly, loss of Gtr1-Gtr2 or GATOR1 ('*iml1Δ*') resulted in delayed starvation response (C, E), while the *iml1Δ* defect was complemented by expressing the GDP-locked mutant ('*gtr1SN*') form of Gtr1 (D, F). The *gtr1Δ* and *iml1Δ* mutations delayed the starvation response even in the *tsc2Δ* (G, H) and *rhb1Δ* (J) backgrounds, suggesting that Gtr1-Gtr2 and GATOR1 negatively regulate TORC1 signaling independently of the TSC-Rhb1 pathway. (K) TORC1-dependent phosphorylation of Sck1 is augmented in the absence of Gtr1$^{GDP}$-Gtr2. The *sck1:FLAG* strains carrying the indicated mutations are grown in EMM at 30°C and their lysate was analyzed by immunoblotting. As a negative control with inactive TORC1, a wild-type strain expressing Sck1-FLAG was starved for nitrogen for 1 hr ('-N'). The band intensity of the phosphorylated (filled arrowhead) and unphosphorylated (open arrowhead) forms of Sck1-FLAG was quantified, and the percentages of the phosphorylated form to the total Sck1-FLAG level in each sample are presented in the bar graph as means ± SD (n = 4 independent experiments). *p<0.01; n.s., not significant, compared to the wild-type control using Student's *t*-test.

DOI: https://doi.org/10.7554/eLife.30880.015

The following source data is available for figure 6:

**Source data 1.** Source data for Figure 6K.

DOI: https://doi.org/10.7554/eLife.30880.016

---

*any1Δ* (*Figure 7G*) mutations restored the plasma membrane localization of Cat1 in the *gtr1Δ* and *iml1Δ* mutants, consistent with the observed genetic complementation (*Figure 7D*).

Taken together, these results strongly suggest that elevated TORC1 activity in cells lacking GATOR1, Gtr1$^{GDP}$-Gtr2 or Ragulator leads to reduced amino-acid uptake, which partly contributes to their growth defects on amino acids as nitrogen source. Although inactivation of TORC1, such as that by *tor2* mutations, also results in growth defects, the *tor2* phenotypes are not complemented by ammonium in the growth medium (*Figure 7—figure supplement 1F*). In addition, the *tor2* mutant cells are smaller than wild-type cells (*Matsuo et al., 2007*), while the mutants lacking Gtr1$^{GDP}$-Gtr2, GATOR1 or Ragulator do not show such a phenotype (*Figure 7—figure supplement 1G*). Thus, elevated TORC1 activity brings about growth defects that are apparently different from those from TORC1 inactivation.

## Discussion

Like in mammalian cells, the Rheb GTPase Rhb1 is an essential activator of TORC1 in fission yeast (*Mach et al., 2000*; *Urano et al., 2005*; *Uritani et al., 2006*). In this study, we have identified and characterized the *S. pombe* counterparts of Ragulator and GATOR1, the mammalian protein complexes that control the Rheb-dependent activation of mTORC1 on the lysosomal surface (*Sancak et al., 2010*; *Bar-Peled et al., 2013*). Because the Rag-like heterodimeric Gtr1-Gtr2 GTPases are also implicated in the regulation of *S. pombe* TORC1 (*Valbuena et al., 2012*), we delved into the roles and relationships of these conserved regulatory factors in the control of TORC1 signaling.

Affinity-purification of the Gtr1-Gtr2 heterodimer followed by mass-spectrometry led to identification of four co-purified proteins, Lam1, Lam2, Lam3 and Lam4 (*Figure 1*). The following observations strongly suggest that the four Lam proteins form a complex equivalent to mammalian Ragulator that recruits the Rag GTPase heterodimer to lysosomes. First, among the components of the Lam protein complex, Lam2 shows detectable sequence similarity to the Ragulator subunit LAM-TOR2/p14 (*Ma et al., 2016*). Second, the predicted secondary structure of Lam3 suggests it has a roadblock domain, which has been found among the Ragulator components. Third, the lipid modification sites and the lysosome/vacuole-targeting di-leucine motif at the N-termini of mammalian LAMTOR1 and Ego1 in budding yeast are also conserved in the *S. pombe* Lam1 protein (*Figure 2*). Lastly, the Lam protein complex is localized to the membranes of vacuoles, lysosome-like organelles in yeast and is essential for the vacuolar localization of the Gtr1-Gtr2 GTPase heterodimer (*Figures 2* and *3*, *Figure 3—figure supplement 1*).

In contrast to the limited sequence similarity of the Ragulator subunits between mammals and fission yeast, the primary structures of the GATOR1 subunits are relatively well conserved through evolution; Iml1, Npr2 and Npr3 in fission yeast are orthologous to human DEPDC5, Nprl2 and Nprl3, respectively (*Figure 4—figure supplements 1* ; *Ma et al., 2013*) and the three proteins indeed form a complex (*Figure 4—figure supplement 2A–C*). Furthermore, the *iml1Δ*, *npr2Δ* and *npr3Δ* defects are complemented by expression of the GDP-locked Gtr1 (*Figures 5F* and *6F*), consistent with the

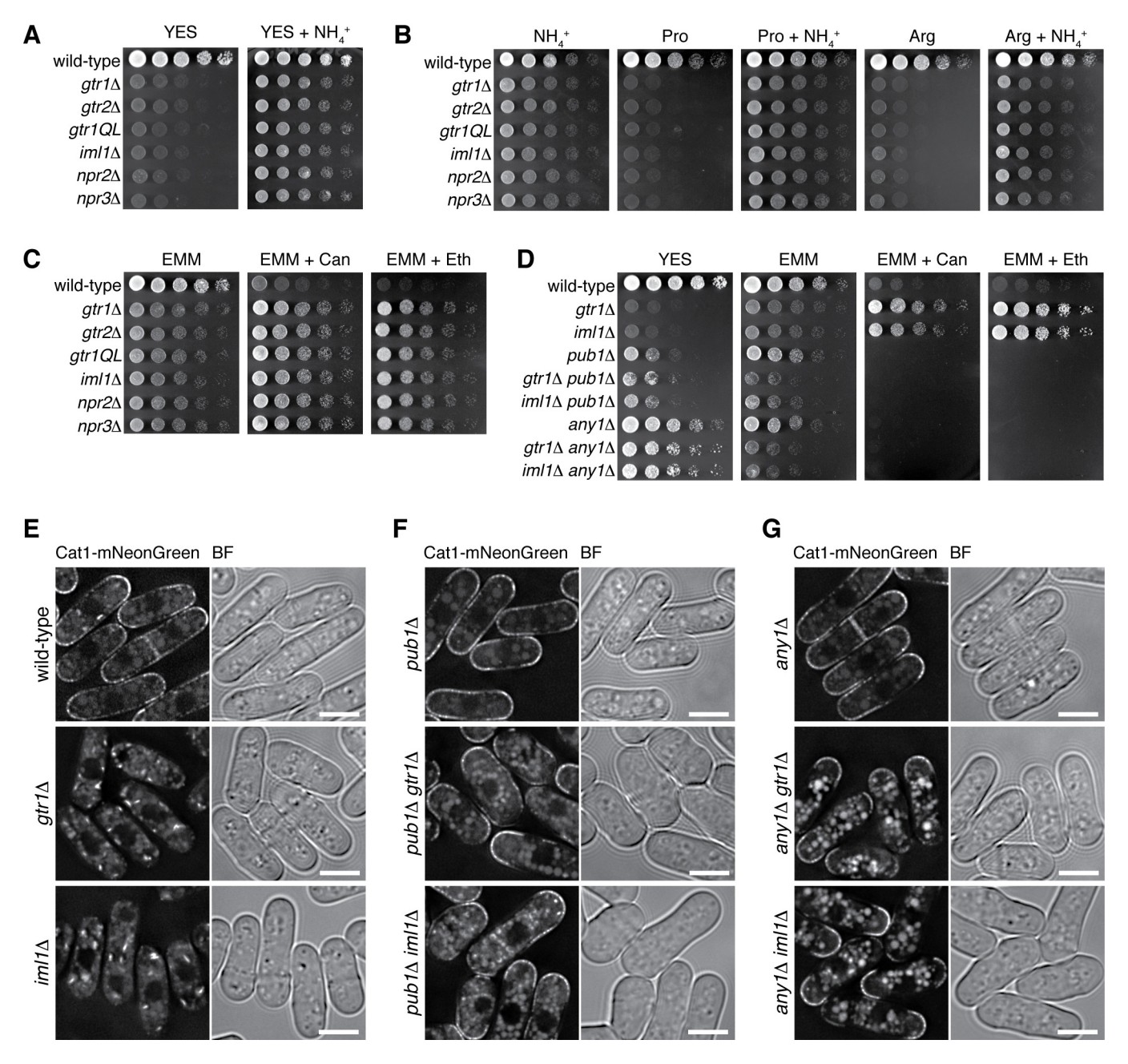

**Figure 7.** Loss of functional Gtr1$^{GDP}$-Gtr2 results in a defect in amino acid uptake. (**A–D**) The indicated strains were grown in EMM and their serial dilutions were spotted onto different agar media at 30°C. The growth defects of the strains lacking Gtr1-Gtr2 or GATOR1 on yeast extract medium (YES) were partially complemented by supplementing the medium with 5 mg/ml NH$_4$Cl (**A**). The mutants also showed severe growth defects on minimal media containing amino acid (20 mM Pro or Arg) as sole nitrogen source, while the phenotype was ameliorated by adding 5 mg/ml NH$_4$Cl (**B**). In comparison to wild-type cells, those mutants were more resistant to canavanine ('Can', 60 μg/ml) and ethionine ('Eth', 30 μg/ml), toxic analogs of arginine and methionine, respectively (**C**), but the phenotype was reversed by inactivating the Pub1-Any1 ubiquitin ligase complex, which promotes internalization of amino acid transporters (**D**). (**E–G**) Translocation of the amino acid transporter Cat1 from the plasma membrane to the cytoplasm in the absence of Gtr1-Gtr2 or GATOR1. The chromosomal *cat1*$^+$ gene was tagged with the mNeonGreen sequence in the indicated strains for microscopic analysis. The *pub1Δ* and *any1Δ* mutations restored the plasma membrane localization of Cat1 in the *gtr1Δ* and *iml1Δ* strains. Bars, 5 μm.

DOI: https://doi.org/10.7554/eLife.30880.017

The following source data and figure supplements are available for figure 7:

**Figure supplement 1.** Loss of Gtr1$^{GDP}$-Gtr2 or Ragulator results in a defect in amino-acid uptake.

DOI: https://doi.org/10.7554/eLife.30880.018

*Figure 7 continued on next page*

*Figure 7 continued*

**Figure supplement 1—source data 1.** Source data for *Figure 7—figure supplement 1G*.
DOI: https://doi.org/10.7554/eLife.30880.019

idea that the GATOR1 function as GAP for the RagA/B GTPases is conserved in *S. pombe*, as has also been found in the budding yeast *Saccharomyces cerevisiae* (*Panchaud et al., 2013a*). On the contrary, *Ma et al. (2016)* recently reported that expression of Gtr1$^{S20L}$, their assumed GDP-locked mutant of Gtr1, failed to complement the *npr2Δ* and *npr3Δ* defects, concluding that those mutations do not alter the nucleotide-bound state of Gtr1 (*Ma et al., 2016*); however, the Ser-20 substitution has never been characterized in any Rag family GTPases (*Figure 5—source data 1*) and may not behave as intended. Their study also proposed the function of Npr2 and Npr3 in tethering Gtr1-Gtr2 to vacuoles, based on the result that the expression and vacuolar localization of Gtr1-GFP and Gtr2-GFP were compromised when either of them was ectopically overexpressed in the *npr2Δ* and *npr3Δ* mutants (*Ma et al., 2016*). Unbalanced expression of the heterodimer constituents may explain the observations, as the GATOR1 mutations did not affect the protein levels and vacuolar localization of GFP-fused Gtr1 and Gtr2 that were expressed from their chromosomal loci (*Figure 4E*, *Figure 4—figure supplement 2F–H*). Thus, the vacuolar localization of GATOR1 appears to be independent of the Ragulator-Gtr complex, and may be mediated by an additional mechanism (*Peng et al., 2017*; *Wolfson et al., 2017*).

Our gene deletion experiments found that the null mutants of the Gtr1-Gtr2 GTPases, Ragulator (Lam1, Lam2, Lam3 and Lam4) and GATOR1 (Iml1, Npr2 and Npr3) exhibit very similar growth defects, which are rescued by the TORC1 inhibitors (*Figure 5G*, *Figure 5—figure supplement 1A*). The double mutant analyses confirmed that they all belong to the same epistasis group, consistent with the idea that both Ragulator and GATOR1 complexes function through the Gtr1-Gtr2 hetero-dimer. Together with other data presented in this paper, we propose that the Gtr1$^{GDP}$-Gtr2 hetero-dimer tethered to the vacuolar surface by the Ragulator complex plays a role in moderating TORC1 activation induced by the Rhb1 GTPase. Although TORC1 activity is essential for cell viability in fission yeast (*Álvarez and Moreno, 2006*; *Matsuo et al., 2007*; *Weisman and Choder, 2001*), its attenuation by Gtr1$^{GDP}$-Gtr2 enables appropriate response to limitation in nitrogen source, allowing localization of the amino acid transporters to the plasma membrane (*Figures 6* and *7*). It is currently unknown how Gtr1$^{GDP}$-Gtr2 counteracts the Rhb1-dependent activation of TORC1. In amino-acid-starved mammalian cells, Tsc2 is recruited to lysosomal membranes by the Rag GTPases and inhibits Rheb, resulting in inactivation of mTORC1 (*Demetriades et al., 2014*). However, such a Tsc2-depen-dent mechanism cannot explain the observations that the effects of the *gtr1Δ* and *tsc2Δ* mutations are additive (*Figures 5L* and *6G*) and that the Gtr1-dependent attenuation of TORC1 appears to be in effect even in the absence of Rhb1 (*Figure 6J*).

It remains possible that GTP-bound Gtr1 is capable of promoting TORC1 activation, as has been found for GTP-bound RagA/B in higher eukaryotes (*Kim et al., 2008*; *Sancak et al., 2008*; *Sancak et al., 2010*). However, such functionality of Gtr1$^{GTP}$ would be cryptic in *S. pombe*, where TORC1 can be strongly activated in the *gtr1* null mutant as well as in strains expressing mostly Gtr1$^{GTP}$, such as *gtr1Q61L* and GATOR1 mutants (*Figure 6*). In this organism, TORC1 localization to vacuolar membranes is not dependent on the Ragulator-Gtr complex (*Figure 3C*), suggesting Gtr-independent targeting of TORC1 to the vacuolar surface where the Rhb1 GTPase activates TORC1. Rag-independent recruitment of mTORC1 to lysosomal membranes has also been reported (*Jewell et al., 2015*), and there might be a conserved mechanism for such TORC1 localization. Moreover, it is conceivable that GDP-bound RagA/B also plays a role in attenuation of mTORC1 sig-naling, a possibility compatible with the observed hyperactivation of mTORC1 in the absence of functional GATOR1 on lysosomes (*Bar-Peled et al., 2013*; *Peng et al., 2017*; *Wolfson et al., 2017*). Because deregulated mTORC1 activation is implicated in cancerous cell proliferation and neurologi-cal disorders including epilepsy (*Huang and Manning, 2008*; *Baldassari et al., 2016*), the molecular mechanisms that negatively regulate mTORC1 warrant further investigation.

## Materials and methods

### Fission yeast strains and general techniques

*S. pombe* strains used in this study are listed in the *Supplementary File 1*. Growth media and genetic manipulations for *S. pombe* have been described previously (*Moreno et al., 1991*; *Shiozaki and Russell, 1997*; *Bähler et al., 1998*). *S. pombe* cells were grown in yeast extract medium YES and synthetic minimal medium EMM. For nitrogen starvation experiments, cells exponentially growing in EMM liquid medium were filtered onto 0.45 µm mixed cellulose ester membrane (Advantec, Japan) and resuspended in liquid EMM lacking ammonium chloride (EMM–N). More than two biological replicates were tested for each experiment. For cell size determination, EMM liquid cultures from three biological replicates were diluted 500 times with the Cellpack Reagent (Sysmex, Japan), and cell volumes were measured using the Particle Analyzer CDA-1000 (Sysmex).

### Spot test assay

For spot test assay, EMM liquid cultures were adjusted to cell density equivalent to $OD_{600}$ 1.0. Serial dilutions of the cells were spotted onto solid media. Images were captured by the LAS-4000 system (Fujifilm, Japan).

### Immunoblotting

Cells were harvested in 10% trichloroacetic acid (TCA) and crude cell lysates were prepared as described previously (*Tatebe and Shiozaki, 2003*). Protein concentration was determined by the Bio-Rad protein assay kit (Bio-Rad, Hercules, California). Proteins were resolved by SDS-PAGE, transferred to nitrocellulose, and probed with antibodies as follows: anti-phospho-p70 S6K (Cell Signaling Technology, Danvers, Massachusetts), anti-Psk1 (custom antibody raised in rabbits against the peptide 'SDDEIAEEGYDFEELEAS') (Sigma, St. Louis, Missouri), anti-Spc1 (*Tatebe and Shiozaki, 2003*), anti-FLAG (M2, Sigma), anti-*myc* (9E10, Covance, Princeton, New Jersey; A-14, Santa Cruz Biotechnolgy, Dallas, Texas), and anti-GFP (Roche, Switzerland).

### Immunoprecipitation and mass spectrometry

For immunoprecipitation, cells were grown to exponential phase in YES and filtered onto 0.45 µm mixed cellulose ester membrane (Advantec). The cells were disrupted in lysis buffer composed of 20 mM HEPES-KOH (pH 7.5), 150 mM potassium glutamate, 10% glycerol, 0.25% Tween-20, 10 mM sodium fluoride, 10 mM p-nitrophenylphosphate, 10 mM sodium pyrophosphate, 10 mM β-glycerophosphate, and 0.1 mM sodium orthovanadate containing PMSF, leupeptin, and protease inhibitor cocktail (Sigma) with glass beads using Multi-beads Shocker (Yasui Kikai, Japan). Anti-FLAG magnetic beads (Sigma) and anti-*myc* affinity gel (Sigma) were used to immunoprecipitate FLAG- and *myc*-tagged proteins, respectively. For serial immunoprecipitation, cell lysates were first subjected to anti-FLAG immunoprecipitation, followed by elution using 3X FLAG peptides (Sigma) and anti-*myc* immunoprecipitation.

Samples to be analyzed by mass spectrometry were first resolved in a 12% Mini-PROTEAN TGX precast gel (Bio-Rad). Each lane was sliced into four pieces and digested with trypsin. Mass spectrometric analysis was performed using the LTQ-Orbitrap XL-HTC-PAL system. MS/MS spectra were analyzed by Mascot server (Matrix Science) and compared against NCBInr protein database (Taxonomy: *S. pombe*).

### Microscopy

Fluorescence microscopic analysis was carried out using DeltaVision Elite Microscopy System (GE Healthcare, United Kingdom). Cells grown exponentially in EMM liquid were stained with FM4-64 dye (Invitrogen, Carlsbad, California; Biotium, Fremont, California) for vacuole visualization. Z-axial images were taken at 0.4 µm with a 100X objective lens. Deconvolution of images was performed using DeltaVision SoftWoRx software. More than 200 cells were analyzed in each microscopy experiment.

## Acknowledgements

We thank M Uritani, T Matsumoto, and the NBRP Japan for reagents. We also thank T Toda and H Tatebe for insightful comments and L Takeuchi, A Hishinuma, and R Kurata for technical assistance.

## Additional information

### Funding

| Funder | Grant reference number | Author |
| --- | --- | --- |
| Japan Society for the Promotion of Science | 26840069 | Tomoyuki Fukuda |
| Suzuken Memorial Foundation | | Tomoyuki Fukuda |
| Japan Society for the Promotion of Science | 17K07330 | Tomoyuki Fukuda |
| Japan Society for the Promotion of Science | 26291024 | Kazuhiro Shiozaki |
| Ministry of Education, Culture, Sports, Science, and Technology | Graduate Student Scholarship | Kim Hou Chia |
| Panasonic Corporation | Graduate Student Scholarship | Fajar Sofyantoro |

The funders had no role in study design, data collection and interpretation, or the decision to submit the work for publication.

### Author contributions

Kim Hou Chia, Formal analysis, Validation, Investigation, Visualization, Writing—original draft, Writing—review and editing; Tomoyuki Fukuda, Conceptualization, Formal analysis, Supervision, Funding acquisition, Investigation, Visualization, Writing—original draft, Writing—review and editing; Fajar Sofyantoro, Formal analysis, Validation, Investigation; Takato Matsuda, Takamitsu Amai, Validation, Investigation; Kazuhiro Shiozaki, Conceptualization, Formal analysis, Supervision, Funding acquisition, Visualization, Project administration, Writing—review and editing

### Author ORCIDs

Kim Hou Chia (iD) https://orcid.org/0000-0002-7958-6635
Tomoyuki Fukuda (iD) http://orcid.org/0000-0003-2069-7127
Kazuhiro Shiozaki (iD) http://orcid.org/0000-0002-0395-5457

### Decision letter and Author response

Decision letter https://doi.org/10.7554/eLife.30880.023
Author response https://doi.org/10.7554/eLife.30880.024

## Additional files

### Supplementary files

• Supplementary File 1. List of key resources used in this study.
DOI: https://doi.org/10.7554/eLife.30880.020

• Transparent reporting form
DOI: https://doi.org/10.7554/eLife.30880.021

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
