## [Decision Letter]

Thank you for submitting your article "GDP-bound form of the Rag family GTPase Gtr1 counteracts the Rheb-dependent activation of TOR complex 1" for consideration by *eLife*. Your article has been favorably evaluated by Philip Cole (Senior Editor) and three reviewers, one of whom is a member of our Board of Reviewing Editors.

The reviewers have discussed the reviews with one another and the Reviewing Editor has drafted this decision to help you prepare a revised submission.

Summary:

Mammalian cell studies have led to a model in which TORC1 activation by amino acids is triggered by its recruitment to the RagA/B GTPase heterodimer that is anchored to lysosomal membranes by the multi-subunit Ragulator complex, which functions as a guanine nucleotide exchange factor (GEF). These activities are counteracted by GATOR, which has GAP activity toward RagA/B to cause the release of TORC1 from lysosomal membranes. Chia et al. identify the Ragulator-like complex in fission yeast and show that it tethers the Gtr1-Gtr2 Rag heterodimer to vacuolar membranes. However, in contrast to mammals, TORC1 localizes to vacuolar membranes independently of the Ragulator-Rag complex. Moreover, Chia et al. report that Gtr1-GDP acts to suppress TORC1 activity.

This is an interesting and well-executed study of the Rag pathway in fission yeast that provides important new insights relevant to the evolution and function of this system of regulation and amino acid sensing at the vacuole/lysosome. The difference in Rag function between mammals and fission yeast is intriguing and suggests that adaptation has occurred in each organism to meet the regulatory needs for the control of TORC1 activity.

Essential revisions:

1) The title of the paper should be changed to better reflect the study. First, the title should note that the analysis was performed with fission yeast (or *S. pombe*). Second, while Gtr1-GDP is identified as a suppressor of TORC1 activity, the link to Rheb is not fully established and should not be stated in the title. The title should reflect the more balanced approach to stating conclusions employed by the authors in the Abstract.

2) "Growth" in all of the genetic experiments in the study is classified as a single cellular process/phenotype. Are growth defects from too much or too little TORC1 activity really the same (proliferation effects, cell cycle, cell size)? Given that TORC1 is established to promote cell growth, rather than impair it, these cellular phenotypes are worth a more careful analysis.

3) The presumptive effects on amino acid uptake in the final section of the study are interesting and might provide insight into the growth defect of the mutants analyzed. The authors should directly test whether elevated TORC1 signaling causes a defect in the uptake or utilization of exogenous amino acids in the context of these mutants.

4) Figure 7. Is the rescue by ammonium due to restoration of nitrogen or attenuation of TORC1?

---

## [Author Response]

Essential revisions:1) The title of the paper should be changed to better reflect the study. First, the title should note that the analysis was performed with fission yeast (or S. pombe). Second, while Gtr1-GDP is identified as a suppressor of TORC1 activity, the link to Rheb is not fully established and should not be stated in the title. The title should reflect the more balanced approach to stating conclusions employed by the authors in the Abstract.

Following the above suggestion, the title of the paper has been changed to “Ragulator and GATOR1 complexes promote fission yeast growth by attenuating TOR complex 1 through Rag GTPases”.

2) "Growth" in all of the genetic experiments in the study is classified as a single cellular process/phenotype. Are growth defects from too much or too little TORC1 activity really the same (proliferation effects, cell cycle, cell size)? Given that TORC1 is established to promote cell growth, rather than impair it, these cellular phenotypes are worth a more careful analysis.

Not surprisingly, the growth defects from too much or too little TORC1 activity are quite different. As reported previously (Matsuo et al., 2007, cited), inactivation of TORC1, such as that in *tor2* mutants, results in phenotypes indistinguishable from those induced by nitrogen starvation, even in the presence of rich nitrogen source. Therefore, the *tor2* mutant phenotypes are not suppressed even by a preferred nitrogen source, ammonium, in contrast to the growth defects from too much TORC1 activity in the *gtr1∆* and *iml1∆* mutants (new Figure 7—figure supplement 1). In addition, like nitrogen-starved cells, *tor2* mutant cells with too little TORC1 activity are significantly smaller (Matsuo et al., 2007), while the mutants with too much TORC1 activity, such as gtr1, iml1 and lam1 mutants, are not any smaller than wild-type cells (Figure 3, Figure 4, Figure 7; Figure 2—figure supplement 1, Figure 2—figure supplement 2; Figure 3—figure supplement 1; Figure 4—figure supplement 2; new Figure 7—figure supplement 1).

3) The presumptive effects on amino acid uptake in the final section of the study are interesting and might provide insight into the growth defect of the mutants analyzed. The authors should directly test whether elevated TORC1 signaling causes a defect in the uptake or utilization of exogenous amino acids in the context of these mutants.

New experiments (Figure 7—figure supplement 1) have been included in the revised manuscript to demonstrate that the *gtr1∆* and *iml1∆* mutants with elevated TORC1 signaling have a defect in the utilization of exogenous amino acids. When these mutants carry mutations in the arginine (*arg3-D4*) or leucine (*leu1-32*) biosynthesis pathways, they fail to grow on the growth media supplemented with arginine/leucine, confirming that the mutants are defective in utilizing exogenous amino acids. These results are also consistent with the defective amino-acid uptake due to deregulated TORC1 signaling in the tsc1/tsc2 mutants (e.g., van Slegtenhorst et al., 2004; Aspuria and Tamanoi 2008; both cited).

4) Figure 7. Is the rescue by ammonium due to restoration of nitrogen or attenuation of TORC1?

Ammonium is one of the nutrients stimulatory to TORC1 in fission yeast (Nakashima et al. 2012, cited). Although ammonium does not further stimulate TORC1 that is already active in rich YES medium, it does not attenuate TORC1 either, as confirmed by a new experiment added to Figure 7—figure supplement 1. Therefore, it is very likely that the rescue of the mutant phenotypes by ammonium in Figure 7 is due to restoration of nitrogen source.